# Perceptions of Nut Consumption amongst Australian Nutrition and Health Professionals: An Online Survey

**DOI:** 10.3390/nu14081660

**Published:** 2022-04-15

**Authors:** Georgie Tran, Rachel C. Brown, Elizabeth P. Neale

**Affiliations:** 1School of Health & Society, Faculty of Arts, Humanities and Social Sciences, University of Wollongong, Wollongong, NSW 2522, Australia; gtran@uow.edu.au; 2Illawarra Health and Medical Research Institute, University of Wollongong, Wollongong, NSW 2522, Australia; 3Department of Human Nutrition, University of Otago, Dunedin 9016, Otago, New Zealand; rachel.brown@otago.ac.nz; 4School of Medical, Indigenous and Health Sciences, Faculty of Science, Medicine and Health, University of Wollongong, Wollongong, NSW 2522, Australia

**Keywords:** nut consumption, health professionals, perceptions, survey

## Abstract

Habitual nut consumption is associated with reduced risk of chronic diseases; however, consumption levels in Australia are below recommendations. This study examined perceptions and knowledge regarding nut consumption among Australian healthcare professionals and their provision of nut consumption recommendations. A cross-sectional online survey of Australian health professionals was conducted in February–April 2020. Questions in the survey included demographic details, participants’ perceptions of nut consumption, and nut consumption recommendations they may make to clients and patients. A total of 204 health professionals completed the survey, of which 84% were dietitians or nutritionists. Health professionals demonstrated basic nutritional knowledge regarding nut consumption; however, non-dietitians/nutritionists lacked knowledge of long-term benefits of nut consumption. Dietitians/nutritionists were more likely to agree that nuts are healthy and do not cause weight gain when compared to non-dietitians/nutritionists (*p* = 0.021). A total of 63% of health professionals advised at least some of their clients to eat more nuts, and this was higher among dietitians/nutritionists (68%) than non-dietitians/nutritionists (31%). While basic nutritional knowledge regarding nut consumption was observed among all health professionals, there is scope for further education, particularly for non-dietitians/nutritionists, to ensure that nutrition information provided to patients and clients is accurate and reflects the current evidence base.

## 1. Introduction

Nuts are a core food recommended in dietary guidelines globally [1], and evidence has shown that nut consumption is associated with a lower risk of chronic diseases such as cardiovascular disease [2]. However, nut consumption in Australia [3], as in many countries [1], remains lower than recommended. Research suggests that some consumers are unaware of the cardioprotective benefits of nut consumption [4,5], with confusion regarding the effect of nut intake on body weight also reported [4,6].

Addressing the various gaps in knowledge and misinformation relating to nut consumption may assist in increasing the population intake of nuts to recommended levels. Health professionals can play an important role in promoting nut consumption. However, previous research suggests that consumers are not being encouraged by their health professional to consume nuts [5]. A study among those with or at high risk of cardiovascular disease and/or type II diabetes observed that only 4% of participants reported that their doctor recommended including nuts into their diet. This study also noted that 64% of participants agreed that they would regularly include nuts if their doctor suggested they do so [5]. Health professionals are, therefore, in a position to help address this gap in consumer knowledge through patient education. Limited research has examined health professionals’ perceptions of nut consumption, although a New Zealand study exploring the perceptions and knowledge of nuts amongst health professionals identified that over one-fifth of GPs and nurses believed that nut consumption could cause negative health impacts such as raising blood cholesterol and causing weight gain [7].

In order to provide accurate education to patients and encourage nut intake in line with public health recommendations, health professionals themselves must be aware of the benefits of nut consumption. A positive perception of nut consumption may lead to an increased encouragement of daily nut intake from the health professional [7]. Exploring perceptions and identifying any gaps that may exist in health professionals’ knowledge regarding nut intake, as well as observing the nut recommendations given in practice, may assist in developing strategies for the promotion of nut intake.

The aim of this study was to examine the perceptions and knowledge regarding nut consumption among Australian healthcare professionals and the provision of recommendations relating to nut consumption.

## 2. Materials and Methods

The self-administered online survey examined the perceptions and knowledge of nuts among Australian health professionals and practitioners. Ethics approval was obtained from the University of Wollongong Human Research Ethics Committee (HE: 2019/422). All survey responses were anonymous, and informed tacit consent was obtained via the completion of the online survey. Reporting of the survey design and results followed the Checklist for Reporting Results of Internet E-Surveys (CHERRIES) [8]. 

### 2.1. Survey Development

The questions in the survey were initially informed by a similar survey of health professionals conducted in New Zealand, which was assessed for content and face validity at the time it was originally conducted [7]. For the present study, the New Zealand survey questions were updated, including altering questions to better suit the Australian population. The survey was pilot tested with a convenience sample of five health professionals, where face and content validity were examined. To assess the relevance and clarity of the questions, a modified Likert scale [9,10] was utilized during pilot testing. Health professionals could rate each of the five sub-sections of the survey using a matrix scale. The matrix scale ranged from one “very irrelevant” to four “very relevant”. One of the pilot testers with survey design experience reviewed the ease of understanding and interpretation of the survey, visual design, and presentation. Survey questions were then altered accordingly following content and face validity testing. 

Upon clicking the link in the online invitation, the participant information sheet (PIS) was displayed first. The PIS informed the participants about the purpose of the study, including potential risks, burdens, and benefits to the participants. Questions in the survey included demographic details such as age and profession, participants’ perceptions of nut consumption, and their recommendations that they may make regarding nut consumption to clients and patients. The survey included both open and close-ended questions (Appendix A). 

The survey was developed using the Qualtrics survey software [11]. The survey consisted of 43 questions presented across 33 screens. Some questions were coded for conditional display, whereby questions were only displayed when certain responses to previous questions were given. Answers were able to be reviewed and changed before submission if necessary. Answers to all questions were voluntary, with participants given the option to leave questions blank, skip questions, or choose responses such as ‘I don’t know’.

### 2.2. Recruitment of Participants

Eligible participants included health professionals and practitioners located in Australia. A range of health professionals/practitioners were eligible to take part in the survey. This included general practitioners, practice nurses, dietitians, nutritionists, naturopaths, and personal trainers. These health practitioners were selected, as they were considered likely to be giving dietary advice to patients. The choice of health practitioners also aligned with those used in the New Zealand survey, which included GPs, practice nurses, and dietitians [7].

Surveys were advertised to all members of relevant professional organizations. An invitation to participate was distributed via online newsletters of professional organizations. These professional organizations included Dietitians Association (DA), the Nutrition Society of Australia (NSA), the Royal Australian College of General Practitioners (RACGP), the Australian Primary Health Care Nurses Association (APNA), the Australian Naturopathic Practitioners Association (ANPA), and the Australian Institute of Personal Trainers (AIPT). At the time of the survey dissemination, there were approximately 128,000 members from these organizations collectively. It should be noted that some individuals may be a member of multiple organizations; therefore, the exact number of individuals who received the survey invitation cannot be quantified. Invitations to take part in the survey were also circulated via investigators’ professional networks. Individuals who were interested in participating could click on the link in the invitation. Participants could enter the draw to win one of five gift vouchers as an incentive for completing the survey.

### 2.3. Data Analysis

Data were initially exported to Microsoft Excel [12], with SPSS Statistics software (New York, NY, USA) [13] used to conduct analysis where appropriate. Only complete responses were included in the analysis. To determine the internal consistency of Likert scale questions exploring health professionals’ perceptions and knowledge of nuts, responses were coded as follows: strongly agree = 1, agree = 2, neither agree nor disagree = 3, disagree = 4, strongly disagree = 5, and I don’t know = 6, with negatively worded questions reverse coded. Cronbach’s alpha [14] was then calculated to determine internal consistency. Characteristics of study participants were presented as means and standard deviations for continuous variables. Frequencies and percentages were used to present categorical data. For the purpose of comparing results between health professions, due to participant numbers, health professionals were grouped into the following: dietitians/nutritionists (comprising dietitians and nutritionists) and non-dietitians/nutritionists (comprising nurses, naturopaths, medical professionals, researchers, and physiotherapists). Demographic characteristics were compared between health professionals using independent *t*-tests for continuous variables (gender and age) and Fisher’s Exact test and Chi-Squared test for categorical variables (education level and number of years in the profession).

Independent *t*-tests were used to compare the means of perception scores between health professionals, for instance, the perception of the effect of consuming nuts on body weight. Similar to the New Zealand survey [7], in order to allow statistical comparisons between health professionals, responses were scored as follows: strongly agree = 1, agree = 2, neither agree nor disagree = 3, disagree = 4, and strongly disagree = 5, with mean scores then calculated and compared between professions. Where respondents answered, ‘do not know’, these answers were not included when calculating the mean score, and thus when calculating the *p*-value. In this study, ‘do not know’ responses were not statistically analyzed. 

For categorical variables such as the reasons for advising clients to eat more nuts, logistic regression and Chi-Squared test were used to compare between health professionals. Where no responses were recorded in one or more of the groups, no *p*-value was calculated as a comparison could not be made. A *p*-value < 0.05 was considered statistically significant. 

## 3. Results

### 3.1. Participant Demographics

The survey was opened on 25 February 2020 and closed on 6 April 2020. Appendix A outlines the professional organizations the survey was disseminated to and outlines the final participation numbers. A total of 265 health professionals consented to respond to the survey, with 204 of the 265 (77%) complete responses obtained. Questions exploring health professionals’ perceptions and knowledge of nuts were found to have a good level of internal consistency, as determined by a Cronbach’s alpha of 0.711.

Among the survey respondents, dietitians and nutritionists were the largest professional group to participate in the survey, making up 84% of all health professionals completing the survey. Within this professional group, *n* = 162 (95%) identified as being qualified as dietitians. Non-dietitians/nutritionists included practice nurses (*n* = 20), naturopaths (*n* = 8), medical professionals (*n* = 2), health researchers (*n* = 2), and a physiotherapist (*n* = 1). Dietitians/nutritionists differed significantly from non-dietitians/nutritionists in terms of gender, age, level of education, and number of years in the profession (all *p*-values ≤ 0.044) (Table 1). Both health profession groups were predominantly women (over 85%). The mean age was 37.9 years, with non-dietitians/nutritionists on average being approximately 8 years older than dietitians/nutritionists. The majority of respondents held an undergraduate or postgraduate degree (98%), with the remaining 2% of respondents holding a certificate/diploma belonging to the non-dietitians/nutritionists group. A higher proportion of non-dietitians/nutritionists practiced for 20+ years (36%) compared to dietitians/nutritionists (13%). A smaller proportion of non-dietitians/nutritionists practiced for 0–2 years (12%) compared to dietitians/nutritionists (25%).

### 3.2. Perceptions and Knowledge of Nuts and Nut Butters in Health Professionals

The responses from health professionals regarding the perceptions and knowledge of nuts and nut butters are depicted in Table 2. Overall, all health professionals agreed that nuts are healthy, high in protein, high in fat, and filling (at least 76% responded ‘strongly agree’ or ‘agree’ for each).

When mean responses were compared between dietitians/nutritionists and non-dietitians/nutritionists, there were various perceptions where significant differences were observed. Dietitians/nutritionists were more likely to agree that nuts are healthy, are high in fat, and can help lower people’s risk of diabetes, and they were more likely to disagree that nuts are low in energy/calories, are low in vitamins/minerals, are low in fiber, are naturally high in salt/sodium, and can increase people’s total blood cholesterol and risk of CVD compared to non-dietitians/nutritionists (all *p*-values ≤ 0.012). In addition, there were some apparent differences between dietitians/nutritionists and non-dietitians/nutritionists in stating that they “did not know” answers to questions. A higher proportion of non-dietitians/nutritionists were unsure whether nuts were high in antioxidants (12% versus 2% for dietitians/nutritionists versus non-dietitians/nutritionists), selenium (for some nuts) (33% vs. 6%), iron (for some nuts) (12% vs. 9%), or that they could help lower people’s risk of diabetes (15% vs. 5%), compared to dietitians/nutritionists.

Table 3 presents the responses from health professionals regarding the effects of nut consumption on body weight. Dietitians/nutritionists were more likely to agree that nuts are healthy and do not cause weight gain when compared to non-dietitians/nutritionists, and this difference was statistically significant (*p =* 0.021). A statistically significant difference was also observed between the groups regarding the statement that nuts are not healthy as they are high in calories and fat (*p* < 0.001), with dietitians/nutritionists more likely to disagree with this statement compared to non-dietitians/nutritionists.

Overall, a large proportion of health professionals (72%) believed that nuts should be included in the diet daily (Appendix A). No health professional believed that nuts should be limited in the diet, or that nuts should not be included in the diet. A larger proportion of non-dietitians/nutritionists believed that nuts should only be eaten in moderation (40%) compared to dietitians/nutritionists (23%), although this difference was not statistically significant. Respondents were able to enter a free text response regarding this topic, where only four dietitians/nutritionists specified that the current recommendation is 30 g of nuts per day. It should be noted, however, that the question did not specify the need to state the recommended serving size.

### 3.3. Nut and Nut Butter Consumption Recommendations Provided by Health Professionals to Their Patients and Clients

A total of *n* = 193 health professionals reported that they provided dietary advice to their clients. Of this number, 63% of health professionals advised at least some of their clients to eat more nuts. As depicted in Figure 1, a large proportion of dietitians/nutritionists advised some of their clients to eat more nuts (68%), whereas only 31% of non-dietitians/nutritionists did so. A very small proportion of health professionals advised their clients to eat fewer nuts, with only 6% of dietitians/nutritionists and 3% of non-dietitians/nutritionists doing so. A very small proportion of dietitians/nutritionists did not mention eating nuts to their clients at all (0.6%), whereas 34% of non-dietitians/nutritionists did not mention nuts to their clients. As the respondents were able to select multiple statements, it was noted that 26% of health professionals selected both statements, indicating that they advised some clients to eat more nuts and also advised some clients to eat fewer nuts. The proportion of respondents who selected both statements was similar between professions, with 25% of dietitians/nutritionists and 31% of non-dietitians/nutritionists selecting both these statements.

Table 4 presents the responses from health professionals regarding the reasons for advising clients to eat more nuts. The most common reasons given included that nuts were good for health/nutritious (82%) and were a good source of protein (74%) and unsaturated fats (74%). While there was some variation in reasons provided by dietitians/nutritionists versus other health professionals, these did not reach statistical significance. The reasons for advising some clients to eat fewer nuts are also represented in Table 4. The most common reason given was that nuts were high in energy/calories (67%). Dietitians/nutritionists were more likely to select this reason compared to non-dietitians (*p* = 0.003).

### 3.4. Reasons Provided by Clients for Not Eating More Nuts, or for Not Eating Nuts at All

Health professionals stated the most frequently reported reasons by clients for their low or lack of nut intake were that they had dental issues, making it inconvenient/uncomfortable for them to eat nuts (56.8%), they were allergic to nuts (52.3%), eating nuts could cause weight gain (45.8%), and they believed nuts were high in energy/calories (43.9%) (Appendix A). The less common reasons included that nuts were naturally high in salt/sodium (5.2%), there was no supply/nuts were difficult to purchase (3.9%), and eating nuts could increase blood cholesterol (2.6%) and cardiovascular disease (0.6%).

### 3.5. Interest in Further Education and Training Regarding Nuts among Health Professionals

The majority of health professionals (69%) indicated that they were interested in undertaking further education/training regarding nut consumption, with 12% stating that they were not interested, and 19% indicating they were unsure. The most popular forms of training selected were summaries of current scientific evidence (*n* = 127), webinars (*n* = 123), and written education resources (for health professionals) (*n* = 120).

## 4. Discussion

To the best of our knowledge, this is the first study exploring perceptions and knowledge of nut consumption among Australian health professionals. It is reasonable to assume that accurate knowledge and positive perceptions of nut intake will likely encourage health professionals to recommend nut consumption to their patients and clients. Due to the suboptimal intake of nuts observed in Australia [3], and the influence that health professionals may have in encouraging nut consumption in the general public, it is important to understand the current perceptions and knowledge of nuts that health professionals in Australia hold. Research has suggested that consumers are more likely to increase their nut intake if their doctor recommended them to do so [5,14], and this may also be the case for other health professionals. The results of this study indicated that the health professionals included in the survey overall had basic nutritional knowledge about nut consumption. However, more specialized nutrition knowledge, for example, the effects of nut intake on cholesterol and risk of CVD, were lacking among non-dietitians/nutritionists. Furthermore, a higher proportion of dietitians/nutritionists were giving dietary advice around nut consumption compared to non-dietitians/nutritionists, aligning with the role of dietitians/nutritionists as experts in nutrition. This research has identified gaps in knowledge and opportunities for improved education, which can influence recommendations made regarding nut consumption. Results may be applicable to other countries where similar dietary patterns to Australia are observed. 

The health professionals who were targeted in this study were all likely to be in a position where patients or clients could ask for basic dietary advice, with a range of professions included to capture the different sources of nutrition information that consumers may be exposed to. In particular, GPs and nurses were included in the recruitment strategy to allow the comparison to a similar study performed in New Zealand [7], although no GPs completed the current survey. The results of this survey indicated that respondents across different professions were aware of the basic nutrient profile of nuts, including that nuts are healthy, high in protein and fat, and filling. Interestingly, agreement with these perceptions was more prominent in the dietitians/nutritionists group. Reasons recorded for health professionals advising patients to eat more nuts aligned with these perceptions, demonstrating an important link between the perceptions held by health professionals and the nutrition information they provide to their patients. Similar results were observed in New Zealand, where health professionals generally thought of nuts as healthy, high in protein and fat, and filling. It was also observed in New Zealand that agreement to these perceptions was more prominent among dietitians than GPs and practice nurses [7].

Differences in the perceptions of nuts between health professionals observed in this study are likely to reflect differences in nutritional knowledge. Although respondents in the current survey were predominantly dietitians or nutritionists, the comparison of perceptions with other health professionals provides insights into how knowledge and recommendations regarding nut consumption may differ between professions. For instance, dietitians/nutritionists were more likely to associate greater health benefits with nut intake than other health professionals. While dietitians and nutritionists have expertise in nutrition, given that all health professionals are in a position to educate patients, it is beneficial for all professionals to improve their current knowledge and to ensure that accurate nutrition information is provided to patients. For example, in response to whether nuts can increase people’s total blood cholesterol and risk of CVD, significantly more non-dietitians/nutritionists incorrectly believed that nut consumption could increase blood cholesterol and risk of CVD, contradicting the current evidence base for the effect of nut consumption on health outcomes [15,16,17]. Similar results were observed in New Zealand where GPs and practice nurses were twice as likely as dietitians to believe nut consumption could increase blood cholesterol [7]. With only 39% of non-dietitians/nutritionists advising clients to eat more nuts to help decrease risk of CVD, and 44% advising clients to eat more nuts to help lower blood cholesterol, these results indicate an opportunity for educating non-dietitians/nutritionists in this area. Considering that 69% of health professionals indicated that they were interested in undertaking further education/training on nuts, this may be a priority topic to note for future health promotion strategies, to ensure that the nutrition information provided to clients is accurate and reflects the current evidence base.

Confusion surrounding the effects of nut intake on body weight has previously been reported in both consumers and health professionals [4,7,18,19] and may stem from the perception that the high fat content of nuts can lead to excess energy intake and therefore promote obesity [19]. In the present study, significantly more dietitians/nutritionists agreed that nuts are healthy and do not cause weight gain, whereas only 60% of non-dietitians/nutritionists agreed with this statement. Furthermore, when health professionals were asked about the common reasons their clients gave for avoiding or limiting nut intake, one of the most common reasons was that eating nuts could cause weight gain and they were high in energy/calories. These results contrast with the evidence relating to nuts and weight gain, with a recent systematic review demonstrating that nut consumption is not associated with weight gain [20]. These findings indicate the need for improved education, particularly among non-dietitians/nutritionists, regarding regular nut consumption and the effects on body weight and health, to ensure accurate nutrition information is provided to clients. 

Due to the many health benefits that nuts provide, national recommendations currently state that 30 g of nuts should be eaten daily [21,22]. Overall, almost three-quarters of health professionals correctly identified that current dietary recommendations suggest including nuts in the diet daily. However, one-quarter of all health professionals believed that nuts should only be eaten in moderation. This highlights potential confusion regarding regular nut consumption and its effects on health. In Australia, most consumers do not follow dietary guidelines, with difficulty in understanding the guidelines a commonly reported reason for this [21,23]. The lack of concordance between population eating patterns and dietary guidelines, specifically the suboptimal intake of nuts observed in Australia, strongly suggests the need for appropriate education on current dietary recommendations for the general public. As health professionals are in a position to facilitate direct education to patients and clients, the emphasis should also be on appropriate education for health professionals. 

This study also provided insights into the recommendations provided to clients regarding nut intake. Of those health professionals who provided dietary advice to clients, around two-thirds of dietitians/nutritionists advised some of their clients to eat more nuts, and roughly one-third of non-dietitians/nutritionists did so. This could be a reflection on the more diet-focused and nutrition specialized consultations that dietitians/nutritionists have with their patients and clients [24,25]. Overall, only a fairly small number of health professionals discouraged nut consumption, and this could reflect the generally positive perceptions of nut intake held by survey respondents. The most common reasons provided by health professionals for advising clients to eat more nuts include that they are good for health/nutritious, they are a good source of protein and unsaturated fats, and they can help promote satiety (fullness). The attention to these nutrients and acknowledgement of nut intake for satiety may suggest that health professionals are aware of the basic nutrient profile of nuts. It may also suggest that confusion regarding nut consumption is not as evident in these areas, and the focus of education for health professionals should address other topics. 

Given the low levels of nut consumption observed in both Australia and internationally [3,26], it is useful to explore reasons that nut intake may be discouraged by health professionals, as well as the reasons for nut avoidance reported by clients. In the case of dietitians and nutritionists, a common reason for advising some clients to eat fewer nuts was due to the high energy content of nuts. While this may reflect the common role of dietitians in managing obesity [27], these results further emphasize the need for education regarding the effects of nut consumption on body weight. Further, one-third of all health professionals specified that they recommended that clients consume fewer nuts because clients were already consuming an excessive portion. A focus on moderating nut consumption could be a reflection of concern regarding the energy and fat content of nuts highlighted previously in this survey, or it may indicate a lack of awareness of the recommended portion size for nuts, highlighting areas of further education. Encouragingly, no health professionals reported that they advised clients to eat fewer nuts because they are unhealthy, or because regular consumption can increase risk of CVD and blood cholesterol, aligning with the general perceptions of nuts reported previously in this survey and in previous research [7].

Understanding the reasons for nut avoidance reported by clients can provide insights to further encourage nut intake. Over half of the respondents reported that their clients stated they had dental issues, making it inconvenient and uncomfortable to consume nuts. This finding was similar to other countries such as New Zealand, where dentition was a commonly reported barrier to nut consumption [4]. Studies have shown no significant differences in health benefits between consuming different forms of nuts, including nut butters and ground nuts [28,29,30]. This indicates that alternative forms of nuts that may be easier on dentition, such as nut butters, could be recommended to these clients. Another commonly reported barrier included having an allergy to nuts. This was expected to be a common barrier, as allergic reactions can be severe and fatal, and with no cure for allergies, individuals must strictly avoid the allergen to prevent life-threatening consequences [31]. Concern that eating nuts could cause weight gain was also a commonly reported barrier. As previously highlighted, confusion exists around the effects of nut intake on body weight; therefore, it was expected that health professionals would report this reason as a common barrier to regular nut consumption observed in their clients. 

Our study has a number of strengths, including survey development, that used robust methods, including testing for content and face validity to ensure that accurate information was gathered and that the survey could be correctly interpreted by participants. The findings strongly support and give a clear insight into future education strategies for health professionals and the priority areas of concern regarding nuts. However, there were also limitations to consider. The majority of participants were female, limiting the results from being generalized to all health professionals. Results were also limited to members of professional organizations and those in the investigators’ networks. While the survey was disseminated to a range of health professions, only a small number of non-nutrition or dietetic health professionals were recruited. Results may, therefore, not be generalizable to all non-dietetic health professionals. As the survey was self-administered, the results are subject to voluntary response bias, as the survey was likely to attract individuals with an existing interest in nutrition and nuts. This may be demonstrated by the high proportion of dietitians and nutritionists participating in the survey. If this is the case, the gaps in knowledge identified in this survey may be larger than estimated in the broader population of health professionals. Content and face validity were investigated, both in the original New Zealand survey [7] and in the present study, and good internal consistency between questions exploring health professionals’ perceptions and knowledge of nuts was found. However, further validation (for example internal reliability and criteria validation) was not explored, which may impact the quality of the survey’s results. To maintain participant anonymity, potentially identifying details including IP addresses were not collected. As a result, it is possible that multiple responses from the same individual were obtained. 

## 5. Conclusions

Overall, the results of this survey indicated that basic nutrition knowledge regarding nut consumption was observed among all health professionals. However, knowledge relating to specific health benefits of nut intake, such as the effects of nut intake on cholesterol and risk of CVD, were lacking among non-dietitians/nutritionists. Currently, more dietitians/nutritionists gave dietary advice around nut consumption to their clients compared to non-dietitians/nutritionists, aligning with their role as nutrition experts. There are several areas where further education could be provided to health professionals, especially non-dietitians/nutritionists, to ensure that the nutrition information provided to patients and clients is accurate and reflects the current evidence base. These areas include the effects of nut consumption on blood cholesterol, CVD risk, and body weight, basic dietary guidelines, and health benefits of regular nut consumption. Although the results emphasize the need for further education among non-dietitians/nutritionists, further education on the role of nut consumption in a healthy diet for dietitians and nutritionists could greatly supplement existing knowledge and understanding. As evidence strongly supports the many health benefits of regular nut consumption, it is concerning that nut consumption levels in Australia are suboptimal. Given that perceptions of nuts may influence the recommendations that health professionals provide regarding nut consumption to their clients, it is likely that better education and knowledge among health professionals can contribute to improving the population nut intake.

## Figures and Tables

**Figure 1 nutrients-14-01660-f001:**
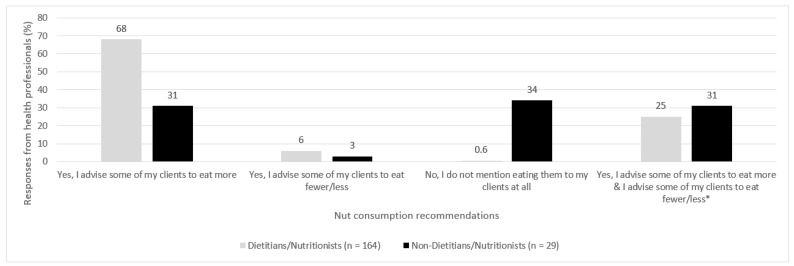
Responses from each health professional group (%) regarding nut and nut butter consumption recommendations to clients. * indicates that participants selected 2 of the available options. These figures are based only on the 193 out of 204 health professionals who answered yes to providing dietary advice to patients or client.

**Table 1 nutrients-14-01660-t001:** Characteristics of study participants.

Demographic	All Health Professionals	Dietitians/Nutritionists	Non-Dietitians/Nutritionists	*p*-Value
*n*	204	171	33	
Female *n* (%)	197 (97)	168 (98)	28 (85)	0.044 ^#^
Age (years)	37.9 (13.4)	36.5 (12.7)	45.3 (14.4)	<0.001 ^#^
**Level of education *n* (%)**				<0.001 *
Certificate/diploma	5 (2)	0 (0)	4 (12)	
Bachelor	85 (42)	67 (39)	18 (55)	
Post-graduate	114 (56)	104 (61)	11 (33)	
**No. of years in the profession *n* (%)**				0.020 ^a^
0–2 years	46 (23)	42 (25)	4 (12)	
3–5 years	36 (18)	32 (19)	4 (12)	
6–10 years	47 (23)	41 (24)	6 (18)	
10–20 years	40 (20)	33 (19)	7 (21)	
20+ years	35 (17)	23 (13)	12 (36)	

All values are means (SD) unless otherwise specified. *p*-values comparing the two health professional groups were from ^#^ independent *t*-test (gender, age), * Fisher’s Exact test (education), and ^a^ Chi-squared test (years in profession).

**Table 2 nutrients-14-01660-t002:** Responses from health professionals *n* (%) regarding the perceptions and knowledge of nuts and nut butters.

	Dietitians/Nutritionists (*n* = 171)							Non-Dietitians/Nutritionists (*n* = 33)							
Statement	Strongly Agree	Agree	Neither	Disagree	Strongly Disagree	Mean	Do Not Know	Strongly Agree	Agree	Neither	Disagree	Strongly Disagree	Mean	Do Not Know	*p*-Value *
**They are healthy**	100 (58)	68 (40)	1 (0.6)	0 (0)	2 (1)	1.46	0 (0)	6 (18)	24 (73)	3 (9)	0 (0)	0 (0)	1.91	0 (0)	<0.001
**They are low in energy/calories**	1 (0.6)	5 (3)	13 (8)	76 (44)	77 (45)	4.30	1 (0.6)	0 (0)	4 (12)	4 (12)	16 (48)	8 (24)	3.87	1 (3)	0.008
**They are high in protein**	40 (23)	108 (63)	20 (11)	6 (3)	1 (0.6)	1.96	0 (0)	7 (21)	19 (58)	5 (15)	2 (6)	0 (0)	2.06	0 (0)	0.463
**They are low in vitamins/minerals**	3 (2)	3 (2)	10 (6)	78 (46)	75 (44)	4.30	2 (1)	0 (0)	2 (6)	6 (18)	18 (55)	5 (15)	3.84	2 (6)	0.004
**They are high in fat**	67 (39)	98 (57)	4 (2)	2 (1)	0 (0)	1.65	0 (0)	4 (12)	21 (64)	6 (18)	2 (6)	0 (0)	2.18	0 (0)	<0.001
**They are low in fiber**	5 (3)	10 (6)	10 (6)	91 (53)	53 (31)	4.05	2 (1)	0 (0)	6 (18)	8 (24)	15 (45)	3 (9)	3.47	1 (3)	<0.001
**They are high in antioxidants**	27 (16)	83 (49)	42 (25)	14 (8)	1 (0.6)	2.28	4 (2)	3 (9)	13 (39)	9 (27)	4 (12)	0 (0)	2.48	4 (12)	0.231
**They are naturally high in salt/sodium**	0 (0)	13 (8)	8 (5)	78 (46)	68 (40)	4.20	4 (2)	0 (0)	7 (21)	4 (12)	20 (61)	1 (3)	3.47	1 (3)	<0.001
**Some of them are high in selenium**	59 (35)	90 (53)	12 (7)	0 (0)	0 (0)	1.71	10 (6)	3 (9)	15 (45)	3 (9)	1 (3)	0 (0)	2.09	11 (33)	0.054
**Some of them are high in iron**	7 (4)	94 (55)	31 (18)	19 (11)	4 (2)	2.48	16 (9)	1 (3)	18 (55)	8 (24)	2 (6)	0 (0)	2.38	4 (12)	0.301
**They are filling**	41 (24)	111 (65)	8 (5)	7 (4)	2 (1)	1.92	2 (1)	6 (18)	22 (67)	4 (12)	1 (3)	0 (0)	2.00	0 (0)	0.583
**Eating them will cause people to gain weight**	0 (0)	9 (5)	55 (32)	78 (46)	28 (16)	3.74	1 (0.6)	0 (0)	5 (15)	9 (27)	15 (45)	4 (12)	3.55	0 (0)	0.222
**Eating them can increase people’s total blood cholesterol**	0 (0)	9 (5)	17 (10)	79 (46)	57 (33)	4.14	9 (5)	0 (0)	4 (12)	10 (30)	14 (42)	3 (9)	3.52	2 (6)	<0.001
**Eating them can increase people’s risk of cardiovascular disease**	1 (0.6)	3 (2)	8 (5)	76 (44)	77 (45)	4.36	6 (4)	0 (0)	5 (15)	7 (21)	13 (39)	6 (18)	3.65	2 (6)	<0.001
**Eating them can help lower people’s risk of diabetes**	21 (12)	90 (53)	44 (26)	5 (3)	2 (1)	2.24	9 (5)	1 (3)	14 (42)	7 (21)	6 (18)	0 (0)	2.64	5 (15)	0.012

Responses scored: strongly agree = 1, agree = 2, neither = 3, disagree = 4, and strongly disagree = 5. * *p*-value for differences between health professionals for mean responses (strongly agree to strongly disagree) was calculated by independent t test. “Do not know” answers were not included in the calculation of the mean.

**Table 3 nutrients-14-01660-t003:** Responses from health professionals *n* (%) regarding the effects of nut consumption on body weight.

	Dietitians/Nutritionists (*n* = 171)							Non-Dietitians/Nutritionists (*n* = 33)							
Statement	Strongly Agree	Somewhat Agree	Neither	Somewhat Disagree	Strongly Disagree	Mean	Do Not Know	Strongly Agree	Somewhat Agree	Neither	Somewhat Disagree	Strongly Disagree	Mean	Do Not Know	*p*-Value
**Nuts are healthy and do not cause weight gain**	42 (25)	85 (50)	31 (18)	13 (8)	0 (0)	2.09	0 (0)	4 (12)	16 (48)	4 (12)	9 (27)	0 (0)	2.55	0 (0)	0.021
**Nuts are healthy but only in moderation**	80 (47)	62 (36)	18 (11)	6 (4)	4 (2)	1.78	1 (0.6)	15 (45)	16 (48)	2 (6)	0 (0)	0 (0)	1.61	0 (0)	0.319
**Nuts are not healthy as they are high in calories and fat**	0 (0)	1 (0.6)	1 (0.6)	50 (29)	119 (70)	4.68	0 (0)	0 (0)	0 (0)	2 (6)	19 (58)	12 (36)	4.30	0 (0)	<0.001
**Eating nuts will cause weight gain**	0 (0)	19 (11)	36 (21)	70 (41)	46 (27)	3.84	0 (0)	0 (0)	5 (15)	6 (18)	15 (45)	7 (21)	3.73	0 (0)	0.549

Responses scored: strongly agree = 1, somewhat agree = 2, neither = 3, somewhat disagree = 4, and strongly disagree = 5. *p*-value for differences between health professionals for mean responses (strongly agree to strongly disagree) was calculated by independent *t*-test. “Do not know” answers were not included in the calculation of the mean.

**Table 4 nutrients-14-01660-t004:** Responses from health professionals *n* (%) regarding the reasons for advising clients to increase or decrease their nut consumption.

Reasons				*p*-Value
**Reasons for advising clients to consume more nuts**	All health professionals (*n* = 171 ^)	Dietitians/nutritionists (*n* = 153)	Non-dietitians/nutritionists (*n* = 18)	
**They are good for health/nutritious**	141 (82)	125 (82)	16 (89)	0.454
**They are a good source of energy/calories**	91 (53)	80 (52)	11 (61)	0.480
**They are a good source of protein**	127 (74)	114 (75)	13 (72)	0.834
**They are a good source of vitamins and minerals**	95 (56)	83 (54)	12 (67)	0.320
**They are a good source of unsaturated fats**	127 (74)	115 (75)	12 (67)	0.438
**They are a good source of fiber**	105 (61)	96 (63)	9 (50)	0.297
**They are a good source of antioxidants**	53 (31)	47 (31)	6 (33)	0.821
**Some of them are a good source of selenium**	56 (33)	47 (31)	9 (50)	0.106
**Some of them are a good source of iron**	42 (25)	35 (23)	7 (39)	0.142
**Eating them can help decrease risk of cardiovascular disease**	93 (54)	86 (56)	7 (39)	0.169
**Eating them can help lower blood cholesterol**	96 (56)	88 (58)	8 (44)	0.294
**Eating them can help promote satiety (fullness)**	124 (73)	109 (71)	15 (83)	0.285
**Eating them can help with weight management**	76 (44)	68 (44)	8 (44)	1.000
**They are a good snack option ***	9 (5)	8 (5)	1 (6)	0.953
**They are a good meat alternative for vegetarians and vegans ***	4 (2)	3 (2)	1 (6)	0.362
**Reasons for advising clients to consume fewer nuts**	All health professionals (*n* = 61 ^^)	Dietitians/nutritionists (*n* = 51)	Non-dietitians (*n* = 10)	*p*-value
**They are unhealthy**	0 (0)	0 (0)	0 (0)	n/a ^#^
**They are high in energy/calories**	41 (67)	39 (76)	2 (20)	0.003
**They are high in fat**	11 (18)	10 (20)	1 (10)	0.479
**They are naturally high in salt/sodium**	2 (3)	2 (4)	0 (0)	n/a ^#^
**Regular consumption of them can increase risk of cardiovascular disease**	0 (0)	0 (0)	0 (0)	n/a ^#^
**Regular consumption of them can increase blood cholesterol**	0 (0)	0 (0)	0 (0)	n/a ^#^
**Regular consumption of them can cause weight gain**	17 (28)	14 (27)	3 (30)	0.869
**There is conflicting information and I do not want to confuse my clients or patients**	1 (2)	0 (0)	1 (10)	n/a ^#^
**There is contraindication(s) with their medication**	1 (2)	1 (2)	0 (0)	n/a ^#^
**They are too expensive for my clients**	6 (10)	5 (10)	1 (10)	0.985
**My clients have dental issues, making it inconvenient/uncomfortable for them**	12 (2)	10 (20)	2 (20)	0.977
**My clients have more pressing concerns than nut consumption**	4 (7)	3 (6)	1 (10)	0.635
**I am concerned about nut allergy**	6 (10)	5 (10)	1 (10)	0.985
**I do not know enough about nuts and their benefits**	1 (2)	1 (2)	0 (0)	n/a ^#^
**My clients consume an excessive portion of them ***	20 (33)	17 (33)	3 (30)	0.837

*p*-value for differences between health professionals was calculated by logistic regression. * indicates free text response that participants submitted and not an option that participants could choose from in the survey. ^ Only 171 out of 204 health professionals advised clients to eat more nuts. Participants could select multiple reasons; therefore, cumulative numbers will not match *n* = 171. ^^ Only 61 out of 204 health professionals advised clients to eat fewer/less nuts. Participants could select multiple reasons; therefore, cumulative numbers will not match *n* = 61. ^#^ indicates no calculated *p*-value, as there were no responses recorded in one or more of the groups.

## Data Availability

The data presented in this study are available on request from the corresponding author.

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
