# Peer review of "Perceptions of Nut Consumption amongst Australian Nutrition and Health Professionals: An Online Survey"

_nutrients, 2022, doi:10.3390/nu14081660_

Round 1
Reviewer 1 Report
The online survey reported by the authors aimed to evaluate the basic nutrition knowledge of health professionals (n= 201, 84% nutritionists) toward nut healthy consumption as well as their intention to recommend them. Considering the general discourse, the low number of respondents but mainly the lack of a standardized survey design (authors used 2004-CHERRIES´guidelines (DOI: 10.2196/jmir.6.3.e34) but not recent guidelines on this matter (DOI: 10.2196/jmir.1923, 10.1371/journal.pmed.1001069)] and validation (internal reliability, criteria validation, etc.; see: 10.1017/S1368980016003372). Based on the latter, this reviewer considers this work as a preliminary analysis that deserves to be completed. Authors are encouraged to consider these arguments, including the unbiased formulation of each question (e.g. the term "good" is not convenient) if possible.
Author Response
The online survey reported by the authors aimed to evaluate the basic nutrition knowledge of health professionals (n= 201, 84% nutritionists) toward nut healthy consumption as well as their intention to recommend them. Considering the general discourse, the low number of respondents but mainly the lack of a standardized survey design (authors used 2004-CHERRIES´guidelines (DOI: 10.2196/jmir.6.3.e34) but not recent guidelines on this matter (DOI: 10.2196/jmir.1923, 10.1371/journal.pmed.1001069)] and validation (internal reliability, criteria validation, etc.; see: 10.1017/S1368980016003372). Based on the latter, this reviewer considers this work as a preliminary analysis that deserves to be completed. Authors are encouraged to consider these arguments, including the unbiased formulation of each question (e.g. the term "good" is not convenient) if possible
Thank you for this feedback. While we agree that there are a range of survey reporting guidelines available, we note that Bennett et al (2011) highlights substantial variation between reporting guidelines. As a result, we feel it is appropriate to follow guidelines specifically designed for surveys conducted via the internet, as we done in this manuscript. We agree that a limitation of this manuscript was the absence of further validation beyond content and face validity. As a result, we have added the following information to the limitations section of the manuscript, and trust this addresses the reviewer’s concerns: “Whilst content and face validity were investigated, further validation (for example internal reliability and criteria validation) were not explored, which may impact on the quality of the survey’s results.”
Reviewer 2 Report
The manuscript is an interesting report about the knowledge of professionals regarding the role of nuts in diets. The evidence from this type of study is relevant in the field, for the understanding of the professionals' educational needs. The methods are well reported, and the paper is clear. I have only a few minor comments for the authors:
- did you control for multiple responses?
- in table 1 there is an alignment problem
- I think Table 2 is really space-consuming. I think there is too much information that could be summarized.
- Have you evaluated the homogeneity of the dietitians/nutritionists group? Because they have a different educational backgroud.
- please revise figure 1 because it is out of the right margin.
Author Response
The manuscript is an interesting report about the knowledge of professionals regarding the role of nuts in diets. The evidence from this type of study is relevant in the field, for the understanding of the professionals' educational needs. The methods are well reported, and the paper is clear. I have only a few minor comments for the authors:
Did you control for multiple responses?
Due to the anonymous nature of the survey, IP addresses were not recorded for responses. As a result, it is possible that multiple responses from the same individual may have been recorded. We have now addressed this as a limitation of this research: “To maintain participant anonymity, potentially identifying details including IP ad-dresses were not collected. As a result, it is possible that multiple responses from the same individual were obtained.”
In table 1 there is an alignment problem
In our version Table 1 appears to be aligned correctly (the p-values for ‘level of education’ and ‘number of years in the profession’ are designed to align with the headings for these categories). If there is an error in the alignment in the reviewer’s version, could the line please be specified, and we will rectify it.
I think Table 2 is really space-consuming. I think there is too much information that could be summarized.
We agree that Table 2 is quite large. However, we feel that the size is justified due to the multiple responses and the need to separate results according to profession. It is hoped that the summary paragraph in the Results section and the discussion of the results can highlight the important statistics
Have you evaluated the homogeneity of the dietitians/nutritionists group? Because they have a different educational backgroud.
We agree that variation may exist with the dietitian and nutritionist group, potentially due to differences in educational background. Exploration of the participants demonstrated that n=162 of the n=171 dietitians/nutritionists identified as dietitians, which suggests a high degree of homogeneity in their qualifications. We have added this information to the results section: “Within this professional group, n=162 (95%) identified as being qualified as dietitians”
Please revise figure 1 because it is out of the right margin.
We have now revised Figure 1 accordingly.
Round 2
Reviewer 1 Report
This reviewer still considers that at least the internal validation of the instrument(survey) could be performed with the collected data (see: https://www.methodspace.com/blog/validating-a-questionnaire, https://www.ncbi.nlm.nih. gov/pmc/articles/PMC5941092/, https://mpra.ub.uni-muenchen.de/103996/1/MPRA_paper_103996.pdf). At least the internal consistency of the questionnaire's items and in-depth discussion of its structure should be added as part of the materials, results, and discussion sections, or as part of supplementary materials to the very least.
Author Response
Thank you for this feedback. We have now assessed the internal consistency of questions assessing participants perceptions and knowledge of nuts using Cronbach’s alpha. We have described this process in the methods and results, and clarified its use in the discussion, as outlined below:
Methods: “To determine the internal consistency of Likert scale questions exploring health professionals’ perceptions and knowledge of nuts, responses were coded as follows: strongly agree = 1, agree = 2, neither agree nor disagree = 3, disagree = 4, strongly disagree = 5, I don’t know = 6, with negatively worded questions reverse coded. Cronbach’s alpha [14] was then calculated to determine internal consistency.”
Results: “Questions exploring health professionals’ perceptions and knowledge of nuts were found to have a good level of internal consistency, as determined by a Cronbach’s alpha of 0.711.”
Discussion: “Content and face validity were investigated both in the original New Zealand survey [7] and in the present study, and good internal consistency between questions exploring health professionals’ perceptions and knowledge of nuts was found. However, further validation (for example criteria validation) were not explored, which may impact on the quality of the survey’s results.”
We have also added further detail on the structure of survey to the methods, which we feel outlines the survey structure in detail, with the full survey available as an appendix: “The questions in the survey were initially informed by a similar survey of health professionals conducted in New Zealand, which was assessed for content and face validity at the time it was originally conducted [7]. For the present study, the New Zealand survey questions were updated, including altering questions to better suit the Australian population”… “Questions in the survey included demographic details such as age and profession, participants’ perceptions of nut consumption, and their recommendations that they may make regarding nut consumption to clients and patients. The survey included both open and close-ended questions, including Likert scales (Supplementary Material 1).
The survey was developed using the Qualtrics survey software [11]. The survey consisted of 43 questions, presented across 33 screens. Some questions were coded for conditional display, whereby questions were only displayed when certain responses to previous questions were given. Answers were able to be reviewed and changed before submission if necessary. Answers to all questions were voluntary, with participants given the option to leave questions blank, skip questions, or choose responses such as ‘I don’t know’”
Reviewer 2 Report
I think the authors have addressed my concerns. I think the manuscript could be considered for publication.
Author Response
Thank you for your feedback
Round 3
Reviewer 1 Report
Thanks